# Achieving tissue-level softness on stretchable electronics through a generalizable soft interlayer design

Yang Li [1], Nan Li[1], Wei Liu [1], Aleksander Prominski[2], Seounghun Kang[1], Yahao Dai[1], Youdi Liu[1], Huawei Hu [1], Shinya Wai[1], Shilei Dai [1], Zhe Cheng[2], Qi Su[1], Ping Cheng[1], Chen Wei[3], Lihua Jin [3], Jeffrey A. Hubbell [1], Bozhi Tian [2] & Sihong Wang [1,4] ✉

Soft and stretchable electronics have emerged as highly promising tools for biomedical diagnosis and biological studies, as they interface intimately with the human body and other biological systems. Most stretchable electronic materials and devices, however, still have Young's moduli orders of magnitude higher than soft bio-tissues, which limit their conformability and long-term biocompatibility. Here, we present a design strategy of soft interlayer for allowing the use of existing stretchable materials of relatively high moduli to versatilely realize stretchable devices with ultralow tissue-level moduli. We have demonstrated stretchable transistor arrays and active-matrix circuits with moduli below 10 kPa—over two orders of magnitude lower than the current state of the art. Benefiting from the increased conformability to irregular and dynamic surfaces, the ultrasoft device created with the soft interlayer design realizes electrophysiological recording on an isolated heart with high adaptability, spatial stability, and minimal influence on ventricle pressure. In vivo biocompatibility tests also demonstrate the benefit of suppressing foreign-body responses for long-term implantation. With its general applicability to diverse materials and devices, this soft-interlayer design overcomes the material-level limitation for imparting tissue-level softness to a variety of bioelectronic devices.

Interfacing electronics with the human body as wearable or implantable devices provides a large collection of functions in health monitoring, disease treatment, and even basic biological studies[1–3]. To achieve seamless and compatible interfaces, it is imperative to minimize the mechanical mismatch between electronic devices and the human skin/tissue, for which the two primary aspects are softness (i.e., Young's modulus) and stretchability[4–7]. In recent years, most of the research efforts have been made in the impartment of stretchability onto electronics[8,9], but with much less progress in reducing the

modulus mismatch with soft tissues[10,11]. Although the achievement of stretchability is often accompanied by the decrease of Young's modulus, the moduli of these stretchable electronic materials (i.e., conductors and semiconductors) and devices are still 3–4 orders of magnitude higher than most soft bio-tissues[12–15] (Fig. 1a). Further reducing such modulus mismatch is highly important for achieving conformable and long-term stable bio-interfaces. First, mechanically, it has been shown, both in theory and in experiments, that a tissue-level modulus can significantly improve the conformability of stretchable

[1]Pritzker School of Molecular Engineering, The University of Chicago, Chicago, IL 60637, USA. [2]Department of Chemistry, The University of Chicago, Chicago, IL 60637, USA. [3]Department of Mechanical and Aerospace Engineering, University of California Los Angeles, Los Angeles, CA 90095, USA. [4]Nanoscience and Technology Division and Center for Molecular Engineering, Argonne National Laboratory, Lemont, IL 60439, USA. ✉e-mail: sihongwang@uchicago.edu

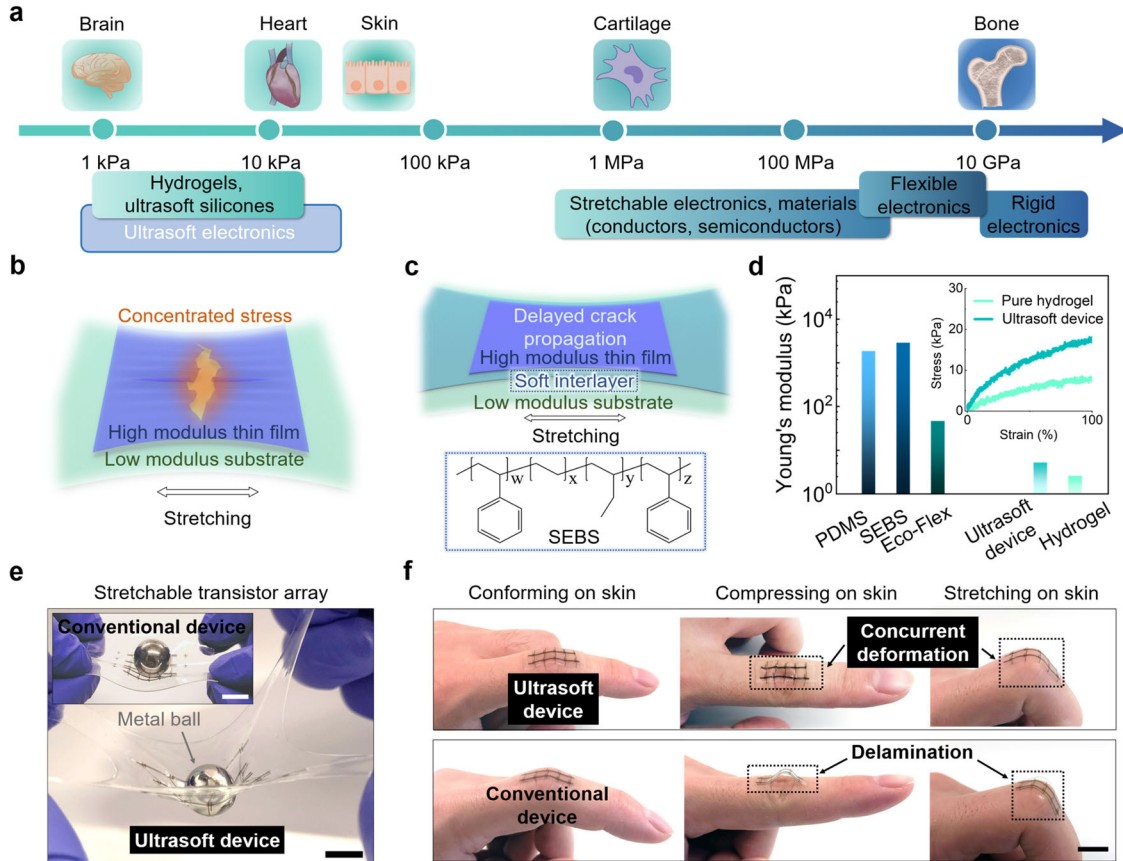

**Fig. 1 | Achieving tissue-like moduli on intrinsically stretchable electronics through soft interlayer design. a** Plotted summary of Young's modulus ranges of the major categories of existing electronics, ultrasoft polymers and hydrogels, and the representative types of bio-tissues. **b** Schematic showing sacrificed stretchability from stress concentration when an electronic thin film is stretched on a substrate with orders-of-magnitude lower modulus. **c** Soft-interlayer design using a thin film with an intermedium modulus to reduce the modulus difference at the interface, which can enable the functional layer to achieve high stretchability. **d** Young's modulus of an ultrasoft and stretchable transistor array built on a PAAm hydrogel substrate, in comparison with the moduli of this hydrogel and several conventional elastomers. The inset shows the strain-stress curves of the ultrasoft transistor array and the hydrogel. **e** Visualization of the much softer property of the stretchable transistor array built on PAAm hydrogel, compared to a reported design of transistor array built on SEBS substrate when under the same deforming force. **f** Highly conformable and imperceptible attachment of an ultrasoft transistor array on a finger joint in different bending positions (upper row), which shows a significant improvement compared to a conventional device on a SEBS substrate of the same thickness (lower row). Scale bars, 10 mm.

devices on nonzero Gaussian surfaces such as the human skin or tissue[16–18]. Second, immunologically, foreign-body responses (FBR) to implantable devices can be significantly suppressed by the tissue-level softness of the bio-implants[19,20]. Collectively, a conformable and scar-free tissue interface is highly important for the fidelity of signal transduction for bio-sensing and modulation[21,22].

To achieve tissue-level moduli, stretchable devices must be built with ultrasoft substrates. However, this is still not readily achievable given the high moduli (i.e., above 100 MPa) of the existing stretchable semiconductors and conductors, because their stretchability, as thin films, will significantly deteriorate on substrates with orders-of-magnitude lower moduli[23,24]. According to the fracture mechanics of thin films supported by elastomeric substrates, a large modulus difference across the interface would aggravate stress concentrations on defect sites and increase the cohesive energy-releasing rate under stretching, thereby creating a higher tendency for crack propagation[25–27] (Fig. 1b). To reduce the modulus mismatch, several ultralow-modulus designs have been created for intrinsically stretchable conductors, either through hydrogel/aerogel designs for PEDOT:PSS[28–32], or dispersion of conductive nanomaterial networks in an ultralow-modulus matrix[10,33–35]. However, tissue-level moduli have never been realized on semiconductors, and thereby active devices[13,36–39] (i.e., the devices with electrical control of charge flow, such as transistors, photodetectors, light-emitting diodes) and signal-

processing circuits, but rather only passive electrodes and passive sensors for signal recording[10,28,31–33].

In this work, we report a generalizable soft-interlayer design (Fig. 1c) that, when added between the functional layer and the substrate, can allow the use of existing stretchable electronic materials to build tissue-level-modulus devices (e.g., transistors, active matrix, sensors), while achieving high stretchability. With such an interlayer simultaneously having an intermediate Young's modulus between the electronic film and the substrate, and sufficient adhesion to both sides, we show the striking result that a very thin (i.e., sub-micrometer) layer can dramatically improve the stretchability of an electronically functional thin film by over a hundred times, as characterized by the electrical performance under 100% strain. Specifically, we find that polystyrene-ethylene-butylene-styrene (SEBS H1052, Fig. 1c) with a Young's modulus of 2.83 MPa can serve as an ideal interlayer, by inherently forming or being enabled with strong adhesions with electrically functional stretchable materials and ultralow-modulus substrates (e.g., soft silicones and hydrogels). The mechanical tests show that the addition of a 1.2 μm-thick SEBS interlayer on a polyacrylamide (PAAm) hydrogel (200 μm-thick) causes a negligible increase in the effective modulus (Fig. 1d), which still stays two to three orders of magnitude lower compared to the commonly used elastomer substrate, such as polydimethylsiloxane (PDMS) and SEBS. Using this soft interlayer design, we demonstrate a stretchable transistor array built

on PAAm hydrogel substrate with an effective Young's modulus of 5.2 kPa, which is two to three order-of-magnitude softer than previously reported stretchable transistor arrays on conventional elastomer substrates (100 μm in thickness) (Fig. 1e). Benefiting from the tissue-like softness, stretchable devices can form much more conformable, stable, and imperceptible contact with dynamic soft surfaces such as the skin with nonzero Gaussian textures (Fig. 1f and Supplementary Fig. 1), as compared to stretchable devices with moduli in the MPa range. It should be noted that although reducing thicknesses without changing the modulus could also help to decrease the stiffness and achieve better conformability, this comes with the price of sacrificing mechanical robustness.

## Results and discussion

### Working mechanism of soft interlayer design

We first theoretically investigate the mechanical functions of the interlayer with varied properties by performing finite-element analysis (FEA) on a three-layer structure composed of a functional film, a soft interlayer, and an ultrasoft substrate. To study the stress concentration, which would facilitate the crack formation and propagation, a notch is introduced in the functional film[40], and the stress level around the notch is mapped (Fig. 2a). With a uniaxial strain applied to the three-layer structure, the stress concentration at the notch tip is studied under the influences of the three following properties of the interlayer: the modulus, thickness, and physical bonding (as reflected by the varied delamination areas) with the functional film (Supplementary Fig. 2). As shown in Fig. 2b–d, a higher modulus, a larger thickness of the interlayer, and less severe delamination (i.e., better adhesion[41,42]) around the notch can all suppress the stress concentration in the functional layer. These effects gradually reach a saturation level, in which the stretchability of the functional layer is dictated by its intrinsic property.

Next, we experimentally verify the soft interlayer design (Fig. 2e) on a stretchable polymer semiconductor created by blending DPPT-TT (poly(2,5-bis(2-octyldodecyl)−3,6-di(thiophen-2-yl)diketopyrrolo[3,4-c]pyrrole-1,4-dione-alt-thieno[3,2-b]thiophene)) with SEBS H1221 in the

weight ratio of 3:7, which has a Young's modulus of 19.4 MPa[43]. The ultrasoft substrate is served by Ecoflex-0010 with Young's modulus of 55 kPa. For the stretchable semiconductor film, 180° peeling tests show that SEBS H1052 (2.83 MPa), compared to other elastomers with similar moduli (i.e., 0.5–3 MPa, Supplementary Fig. 3), has the strongest adhesion (Supplementary Figs. 4, 5). With such an interlayer design, the stretching-induced crack size and density in the semiconductor film under 100% strain indeed get significantly reduced even with only 200-nm thick SEBS interlayer, as compared to stretching on a bare Ecoflex substrate (Supplementary Fig. 6a, b). The further increase of the SEBS layer to 2 μm helps to achieve even less crack propagation (Fig. 2f, and Supplementary Fig. 6c). When characterized in thin-film transistors using the soft-contact lamination method[44] (Fig. 2g), the changing trends of the charge-carrier mobility (Fig. 2h) from the semiconductor film also reflect the markedly improved stretchability given by such SEBS interlayers.

Besides the stretchable polymer semiconductor, the applicability of this interlayer design is also tested on three different stretchable conductor designs: PEDOT:PSS blended with stretchable perfluorinated resin (PFI)[45], carbon nanotube (CNT) assembly[46], and silver nanowire (AgNW) assembly[47], all with Ecoflex-0010 as the substrate (summarized in Supplementary Table 1). Specifically, SEBS still serves as the interlayer for the former two conductors (Supplementary Fig. 7a, b); but thermoplastic polyurethane film has higher Young's modulus than SEBS (also forms strong adhesion with AgNW electrode, Supplementary Fig. 7c), and therefore is a better option[48]. As revealed by the optical microscopy images and the conductivity measurements, the interlayer design also works very effectively for all these three conductors in improving the stretchability (Supplementary Figs. 8–10). According to simulation and experimental results, the three design aspects should meet the following semi-quantitative relationships with the parameters of the functional layers: (1) a modulus within three orders of magnitude to the functional layer; (2) adhesion with an interfacial toughness larger than 100 J m⁻²; (3) thickness at least ten times thicker than the functional layer.

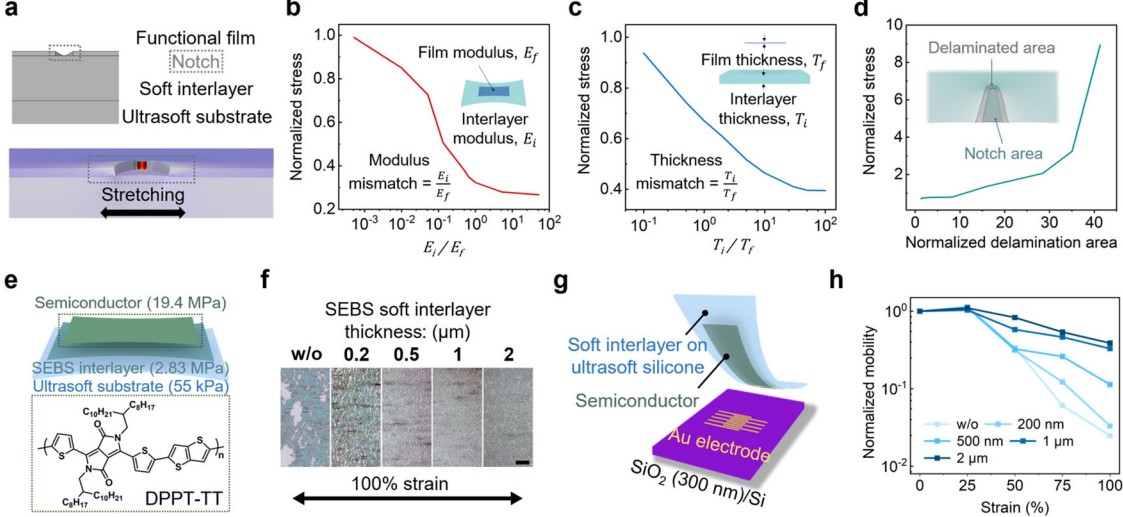

**Fig. 2 | Mechanical effect of the soft interlayer design in significantly improving the stretchability of functional layers on a much softer substrate. a–d** FEA simulation of the suppression of the stress concentration behavior given by a soft interlayer of three design parameters: modulus ($E_f$ and $E_i$ for Young's modulus of film and interlayer, respectively), normalized by that of the functional film (**b**), thickness ($T_f$ and $T_i$ for the thickness of film and interlayer respectively), normalized by that of the functional thin film (**c**), and delamination condition, delaminated area around the notch in the film normalized by notch area (**d**) (stress is normalized by the modulus of the functional film). **e** The soft interlayer design is applied to a stretchable polymer semiconductor film for the integration on an ultrasoft Ecoflex substrate, by using a SEBS film as the interlayer. The moduli of the three layers are labeled in the schematic image. **f** Representative optical images of cracks formed in the stretchable semiconductor films with the SEBS interlayers of varied thicknesses (scale bar 20 μm). **g** Soft contact lamination method for the measurement of these semiconducting films on an ultrasoft substrate during stretching. **h** Representative evolvement of the charge-carrier mobility of these stretchable semiconductor films with different SEBS interlayer thicknesses, during the stretching from 0 to 100% strain.

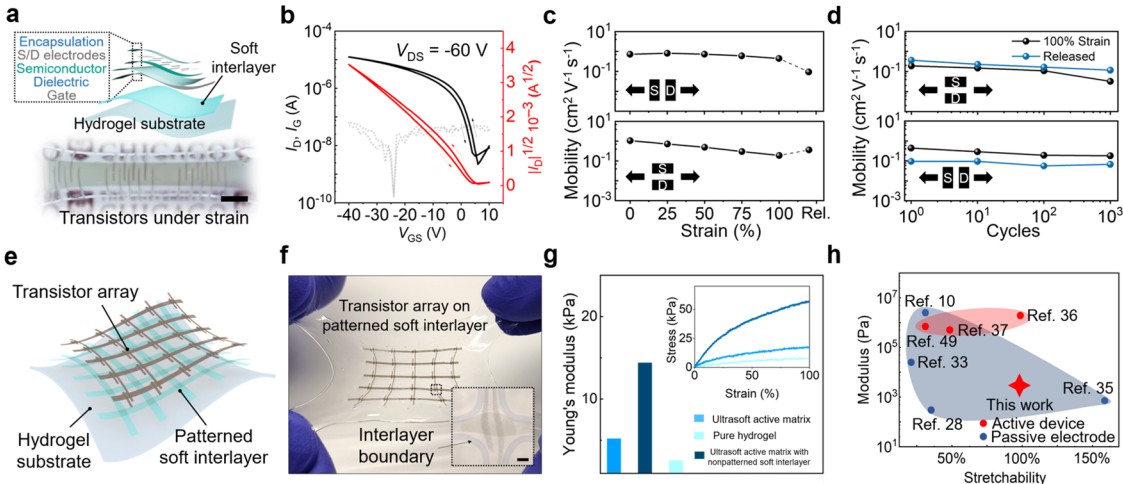

**Fig. 3 | Ultrasoft and stretchable transistor and active matrix circuit built with the soft interlayer design. a** Device structure of the ultrasoft transistors, in which the soft interlayer is simply covered everywhere under the transistors. The inset is an optical image of the transistors under stretching (scale bar, 5 mm). **b** Typical transfer curve of the ultrasoft transistor. **c** Charge-carrier mobility during stretching to 100% strain both in parallel (top) and perpendicular (bottom) to the charge transport direction. **d** Charge-carrier mobility during cyclic stretching to 100% strain in both directions. **e** Structure of the ultrasoft and stretchable active

matrix with a patterned soft interlayer. **f** Representative optical image of an ultrasoft active matrix under biaxial stretching; the inset shows the zoom-in view of one transistor (scale bar: 500 μm). **g** Young's modulus of an ultrasoft active matrix device, in comparison to a similar type of device but with a non-patterned interlayer, which gives higher modulus, and PAAm hydrogel as the substrate.
**h** Comparison of the stretchability and effective modulus of the ultrasoft active matrix in this study to previously reported stretchable active devices and passive electrodes.

## Ultrasoft transistors with soft interlayer design

Next, we use the soft-interlayer design to achieve tissue-like softness on stretchable transistors, which serve as the central type of device for advanced electronics, but with relatively complicated structures. Specifically, the stretchable transistor (Fig. 3a and Supplementary Figs. 11–12) was built with a bottom-gate-top-contact device structure, with SEBS as the dielectric layer, DPPT-TT/SEBS stretchable polymer semiconductor[43] as the channel layer, and CNT network as the gate, drain, and source electrodes. For realizing the ultralow modulus, here we use PAAm hydrogel (200-μm thick, 2.58 kPa in modulus) as the substrate with SEBS soft interlayer, which also provides electrical encapsulation for the device (Supplementary Fig. 13). With this structure, the obtained transistor in a thickness of 200 μm achieves a low Young's modulus of 14.4 kPa. Electrically, the transistor shows ideal transfer behavior with an on/off ratio of $10^4$ (Fig. 3b and Supplementary Fig. 14). During stretching to 100% strain in both directions to the charge transport, the high stretchability facilitated by the soft interlayer ensures stable charge-carrier mobility at the same level ~0.7 cm² V⁻¹ s⁻¹ (Fig. 3c and Supplementary Fig. 15), and even during 1000 repeated stretching cycles at this strain level (Fig. 3d).

For the further use of this design concept to create ultralow-modulus circuits, which generally include multiple interconnected devices, we propose that the soft interlayer can be patterned to only occupy the areas with the functional layers on the top (Fig. 3e). As integrated circuits always have open areas between devices and interconnects, leaving these areas without the coverage of soft interlayer can help the full circuit to have an as-low-as-possible modulus. Here, we apply this on a stretchable active matrix made from transistors with interconnects (Fig. 3f and Supplementary Fig. 16), through laser-engraving patterning of the SEBS interlayer (the fabrication process shown in Supplementary Fig. 17). Such spatially heterogeneous design of the soft interlayer indeed helps the active matrix to achieve an even lower modulus of 5.2 kPa compared to the individual transistors that have the whole-area coverage (non-patterned) of SEBS (Fig. 3g). Functionally, all the transistors fabricated in this ultrasoft active matrix give a highly uniform performance (Supplementary Fig. 18). Overall, compared to all reported stretchable devices (only except for a few passive-electrode sensors, as shown in Fig. 3h and

Supplementary Table 2), we realized the over the 2-order-of-magnitude decrease of the modulus for stretchable active devices while keeping the similar stretchability with the state-of-the-art[36,49]. Therefore, we achieved truly tissue-like mechanical properties in both aspects of softness and stretchability on active electronics.

## Conformability and constraint of ultrasoft devices

The achievement of tissue-level softness can greatly benefit the conformability and the mechanical imperceptibility of electronic devices attached to irregularly curvilinear and dynamic skin/tissue surfaces, which is critical for improving the quality of signal transduction while minimizing discomfort. Although high conformability could be achieved with ultrathin films, modulus mismatch could still largely influence the conformability under deformation[50,51]. To unravel this difference, we quantitatively compare the bonding strength of our ultrasoft device with a conventional stretchable device—both conformably attached to the skin replica made of Ecoflex-0010. When the skin replica is under either stretching or in-plane compression, which introduces shear stress at the interface with the device (Fig. 4a, with the experimental setup shown in Supplementary Fig. 19), the adhesion of the ultrasoft device is much less influenced than the stiffer elastomer device (Fig. 4b). Such improvement in conformability is further proved on nonzero Gaussian surfaces, e.g., human wrist surface as shown in Fig. 4c, d. Our device can much better follow the original skin surface texture under skin deformation, which means better imperception and conformability.

Furthermore, devices of low Young's moduli minimize the mechanical irritation and constraint to the movements of soft skin and tissue. To quantify the different levels of mechanical constraint from soft devices in different moduli, we used digital image correlation (DIC) to map the strain distribution when a skin replica with an attached soft device is under deformation. As shown in Fig. 4e (with the sample preparation shown in Supplementary Fig. 20), under a global strain of 100%, an ultrasoft model device with our interlayer design causes minimal interference to the uniform strain distribution. In comparison, a SEBS-elastomer-based model device (i.e., with a modulus on par with the reported stretchable transistors) distorts the deformation on the skin replica significantly. Similar results are also

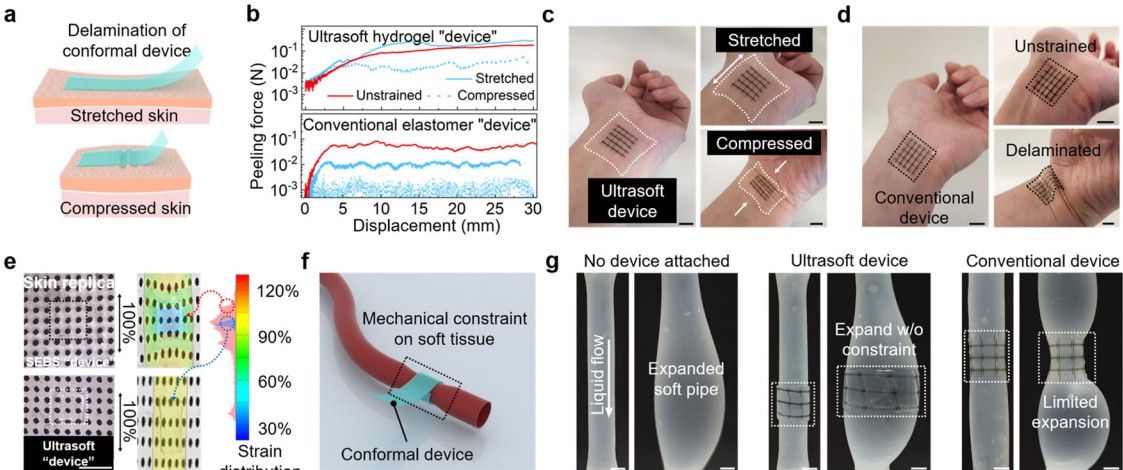

**Fig. 4 | Enhanced conformability and reduced mechanical constraint from the ultrasoft stretchable active matrix when attached onto irregular and deforming surfaces. a** Two scenarios for the testing of the adhesion stability of a model device sheet on a deformed skin surface. **b** 180-degree peeling test results for an ultrasoft model device (i.e., a hydrogel film, 200-μm thick) and a conventional model device (i.e., a SEBS film) on-skin replicas under three different deformation conditions. **c** Visualized conformability of an ultrasoft active matrix attached to a wrist under inward and outward bending, which follows the skin texture much better than a conventional active matrix shown in **d** (scale bars,

10 mm). **e** Digital image correlation (DIC) mapping of strain distribution on a stretched skin replica with the attachment of an ultrasoft and a conventional (i.e., SEBS) model device, respectively, which shows the minimal mechanical constraint from the ultrasoft device (scale bar, 5 mm). **f** Schematics showing a scenario that a soft device that wraps around a deforming organ or tissue (e.g., a blood vessel) in an implantable use. **g** A soft pipe made from Ecoflex emulating a blood vessel with radial expansion: no device attachment (left), wrapped by an ultrasoft device (middle), and by a conventional device (right) (scale bars, 5 mm).

observed for an ultrasoft transistor array (Supplementary Fig. 21). Besides surface attaching, the benefit of low mechanical constraint is even more obvious in the case of wrapping around a deforming soft tissue, e.g., a vessel with periodic expansion (Fig. 4f). This has been demonstrated through a model vessel made of Ecoflex-0010, with pumped liquid mimicking blood flow that causes the vessel expansion (Fig. 4g, left). When wrapped by an ultrasoft transistor array, the vessel can much better keep its original degree of expansion (Fig. 4g, middle), as compared to the significant constraint seen from the SEBS-supported device (Fig. 4g, right).

## Biocompatibility characterization for ultrasoft devices

For implantable applications, basic immunological studies have found that a tissue-level ultralow-modulus tends to produce less FBR[52,53]. However, relevant knowledge is lacking for our design with modulus heterogeneity and gradience, i.e., an ultralow-modulus substrate with a stiffer thin layer at the surface. To investigate this, we compared FBR on three types of samples: a hydrogel film, a SEBS film, and a SEBS-on-hydrogel bilayer sample that emulates our ultrasoft device designs with soft interlayers. These samples were implanted subcutaneously in mice for a period of 1 month to analyze the tissue response. On the explanted tissue samples, Masson's Trichrome staining was applied to reveal the scar tissue (Fig. 5a–c). As shown in Fig. 5d, the scar tissue from the bilayer implant is thinner (i.e., 18.3-μm thick) than the SEBS implant (i.e., 51-μm thick), and similar to the hydrogel implant (i.e., 15.4-μm thick)[54]. These results suggest that long-term immune compatibility could indeed greatly benefit from the decrease of the overall modulus of an implant, possibly due to the reduced pressure and friction caused to the surrounding tissue by the implant[55,56]. This trend is also supported by the analysis of immunofluorescence (α-SMA for collagen and CD68 for macrophage) staining (Supplementary Fig. 22).

To demonstrate the benefit of tissue-like softness on electronic devices with bio-interfacing applications, we further applied our interlayer design on an ultrasoft, stretchable electrophysiological (EP) recording device (Supplementary Figs. 23–24), which was utilized for ex vivo electrocardiography (ECG) recording on an isolated rat heart[49,57]. This device mainly consists of an array of patterned

electrodes based on stretchable PEDOT:PSS composite and a stretchable silicone encapsulation layer (10-μm thick) patterned to cover the electrodes except for the tip areas. Using PAAm hydrogel (200-μm thick) as the substrate together with a SEBS interlayer (1-μm thick), the overall modulus of the device is only 62.9 kPa (Supplementary Fig. 25). The patterning of the electrodes and the encapsulation layer are realized through laser engraving and screen printing (Supplementary Fig. 26). This EP device is attached to the heart surface by wrapping it as a circular ring. Due to the existence of a buffer solution, capillary force formed between the heart and device would provide necessary adhesion (Supplementary Fig. 27). A pressure-sensing balloon is inserted into the left ventricle (LV) (Fig. 5e). During the recording process, the ultralow modulus of the device demonstrated two major benefits. First, the device maintains a relatively stable attachment to the heart surface with minimal positional shifting due to the beating of the heart (Fig. 5f-top, and Supplementary Movie 1), which is highly important for long-term spatial-resolved recording/mapping of ECG on the heart surface (with recorded ECG signals shown in Fig. 5g-top and Supplementary Fig. 28). In comparison, the control device with a SEBS substrate had obvious upward shifting after 5 minutes (Fig. 5f-bottom and Supplementary Movie 2). The influence of such location shifting is reflected in some observable changes in the recorded ECG amplitude (Fig. 5g-bottom and Supplementary Fig. 29). Such stable EP recording has also been achieved on the human skin surface with a similar design (Supplementary Fig. 30). Second, the ultrasoft device causes low mechanical constraint to the beating of the heart as reflected by the stable LVP with minimal increase (Fig. 5h), which is very different from the attachment of a SEBS-based device. This benefit can be important for minimally invasive and stable cardiac mapping using chronically integrated devices.

In summary, we have demonstrated a general soft interlayer design that allows stretchable functional materials with relatively high moduli to be used for creating ultrasoft and stretchable electronics. By achieving a uniform strain distribution, this design can significantly increase the stretchability of a functional layer on a substrate with orders-of-magnitude lower modulus. By allowing stretchable semiconductors to be integrated onto ultrasoft substrates without

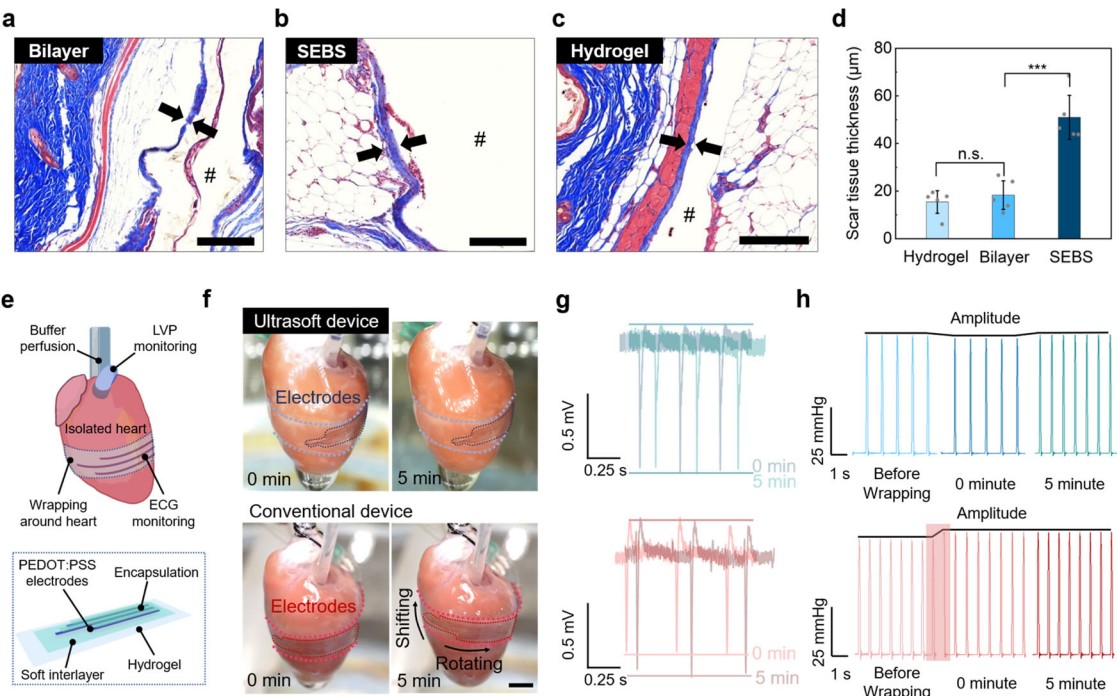

**Fig. 5 | Ultrasoft devices for bio-interfaced applications with suppressed immune responses and mechanical compatibility with living tissues.**
**a**–**c** Images of Masson's trichrome staining for immune responses to three types of model devices after 4-week subcutaneous implantation in mice: an ultrasoft device with a soft interlayer (i.e., SEBS/hydrogel bilayer with the thicknesses of 1.2 μm and 200 μm, respectively), a SEBS device, and a hydrogel device (both in the thickness of 200 μm for the latter two). Implants labeled with #; scale bar, 200 μm. **d** Measured scar tissue thicknesses from the Masson's Trichrome staining images for the three cases. Data for scar tissue thickness are represented as mean values ± standard deviation (s.d.) from five samples, and the corresponding data points are overlaid. (statistical significance was analyzed via two-tailed *T* test, ***\*P* = 0.00032; n.s., not significant). **e** Schematics showing an ultrasoft EP-record device attached to a rat heart for measuring ECG signals. **f** Pictures showing an ultrasoft device wrapping on a beating heart without visible location shift during a period of 5 minutes recording (top). In comparison, an EP device built on a SEBS substrate is observed with some rotation and upward shifting (bottom) (scale bar, 5 mm). **g** ECG signals recorded by an ultrasoft device (top) and a conventional SEBS device (bottom). The maintained recording location of the ultrasoft device on the heart is also reflected by the consistency of the signal features across the 5-min period, which is very different that those from the conventional device. **h** Left ventricle pressure (LVP) recorded when the devices are wrapped on the heart: stable LVP is maintained before and after mounting the ultrasoft device (top), whereas a conventional device with higher mechanical constraint to the heart beating induced an obvious increase in the LVP amplitude (bottom).

compromising stretchability, this interlayer design enabled the realization of stretchable transistors and active matrices with sub-10 kPa moduli, which is over two order-of-magnitude lower than the current state-of-the-art. Through on-skin integration, subcutaneous in vivo implantation, and ex-vivo cardiac recording, we demonstrated that the tissue-like modulus has several advantages, including great conformability to irregular surfaces, low mechanical constraint and irritation for skin and tissues, and improved biocompatibility. The versatility of our soft interlayer design will enable ultrasoft modulus to be achieved on other types of stretchable devices, such as photodetectors, light-emitting diodes, biosensors, and energy-harvesting devices.

## Methods
### Materials
The mechanical buffering interlayer of SEBS H1052 was purchased from *Asahi Kasei*. Other elastomers used in this work were purchased from the corresponding vendors: EcoFlex-0010 (Smooth-on), TPU (BSF), PDMS Sylgard 184 (Dow), and Dragon Skin 35 (Smooth-on). Ultrasoft PAAm hydrogel was prepared from monomer acrylamide (>98%, Sigma-Aldrich); crosslinker N,N'-methylenebisacrylamide (99%, Sigma-Aldrich); thermal initiator ammonium persulfate (>98%, Sigma-Aldrich) and catalyst Tetramethylethylenediamine (99%, Sigma-Aldrich). Stretchable conductors: P3 Carbon nanotube powder (>90%), P2 Carbon nanotube powder (>90%) (Carbon solutions); silver nanowire solution (Kexin Corp. Jiang Su, China). Stretchable polymer conductor: PEDOT: PSS PH 1000 (1.3 wt.% in water, Heraeus); Nafion solution (~5% in water and alcohols, Sigma-Aldrich); Triton-100X (99%,

Sigma-Aldrich) and DMSO (99%, Sigma-Aldrich). All materials are used without further purification. Polymer semiconductor DPPT-TT was synthesized according to the previous report[58].

### Characterization
All measurements were performed in the ambient environment and at room temperature. Resistance measurements for the stretchable electrodes were using the Keithley 6514 with the NI DAQ system. Electrodes were mounted on a uniaxial stretcher, and the resistance was continuously recorded during the stretching. Electrical characteristics of semiconductor thin film and ultrasoft transistors were measured with Keithley 4200 parameter analyzer with a probe station with CLARIUS V1.9. For the tensile test and peeling test of the ultrasoft devices, Zwick-Roell zwickiLine Z0.5 was used with TestXpert II. Strain mapping from digital image correlation was based on optical images obtained from a camera and the software GOM Correlate Pro was used to analyze the strain level. LCR meter (Keysight E4980AL) is used for the impedance measurement of the electrode array.

### Fabrication of ultrasoft transistor and active array devices
Ultrasoft transistors were made based on the transfer printing process from previously reported results[36,43]. A stretchable semiconductor thin film was prepared using polymer solution (10 mg/mL) in chlorobenzene spin-coated on octadecyltrichlorosilane (OTS) treated Si wafer at 1000 rpm for 1 minute followed by annealing in a glove box at 135 °C for 30 minutes[43]. And a stretchable dielectric layer was prepared with 80 mg/mL SEBS H1052 in toluene, which was also spin-coated on

OTS-treated silicon wafer at 1000 rpm for 1 minute. CNT-based gate electrode was spray-coated (1.5 mL/minute) on the oxygen plasma treated (100 W, 5 minutes) silicon dioxide wafer with a concentration of 0.0416 w% (P2-CNT in Methylpyrrolidone, NMP), at the temperature of 200 °C. Soft interlayer was made of 60 mg/mL SEBS H1052 in toluene, and spin-coat on OTS Si wafer for 1000 rpm. Relation between the thickness of SEBS interlayer and spin-coating rate has been summarized in Supplementary Fig. 31. On the top of the interlayer, a Dextran sacrificial layer (5 w% in DI water) was then spin-coated at 2000 rpm after the oxygen plasma treatment for 1 minute (100 W). To start the transfer printing process, a SEBS stamp was used to pick up the interlayer first. Then, followed by transferring the gate electrode. The dielectric layer and semiconductor layer were transferred on the top of the gate electrode using the PDMS stamp successively. S/D electrodes made of P3-CNT (0.0364 w% in 70 mL isopropanol alcohol with three drops of DI water) were then spray-coated on the semiconductor layer through a metal shadow mask (spray rate 3 mL/minute)[36]. The top encapsulation layer made of SEBS H1052 (60 mg/mL, spin-coat on OTS Si wafer at 1000 rpm for 1 minute) was then patterned with a razor blade and transferred using a PDMS stamp to cover the channel area. After all transfer printing processes, the assembled device on a soft interlayer was then fixed with a PET tape-based rigid frame and released from the stamp by soaking in DI water for 30 minutes. Following that, the device was dried in a vacuum hot-plate for 1 hour (80 °C). For the ultrasoft hydrogel substrate, the modulus of the ultrasoft hydrogel was decreased by lowering the crosslinking density. And for the precursor solution, 7.1 g, 100 mmol of acrylamide, 5 mg, 0.032 mmol of crosslinker, 7 mg, 0.031 mmol of initiator, and 27 μL, 0.181 mmol TEMED were dissolved in 50 mL DI water. The precursor solution was then injected into a PET mold with a height of 200 μm to form the hydrogel thin film, which was further heated in an oven at 80 °C for 15 minutes to get fully crosslinked. For the transistor devices without a patterned interlayer, the device supported by a rigid frame was then directly laminated to the surface of the hydrogel thin film. And the hydrogel substrate for the active array device was prepared with enhanced surface adhesion to the array device. Firstly, a very thin layer of SEBS adhesion layer was spin-coated on one side of the PET mold (SEBS H1052 30 mg/mL at 6000 rpm) and followed by forming hydrogel on top of the thin SEBS adhesion layer.

The fabrication of the ultrasoft active array also follows similar film preparation and transfer printing processes for each layer. However, before transfer printing, each layer was separately patterned through a desktop laser machine (OMTech CO$_2$ laser, 40 W). All layers were patterned with the same power of 6.5% with a writing speed of 10 mm/s. And the pattern of each layer is shown in the Supplementary Information (Fig. S16). Differently, the PDMS stamp used for the transfer was treated with oxygen plasma for 15 seconds (100 W), and a heat-up process in an 80 °C oven for 1 minute was required after laminating to the patterned layers. When the active array device was detached from the SEBS stamp and supported on a rigid frame, the device was flipped and then laminated to an OTS-treated glass slide. With the help of align markers on the active matrix, the laser was then used to pattern (6.5 W, 10 mm/s) the soft interlayer. The redundant SEBS is then removed from the glass slide by a tweezer. Finally, a piece of ultrasoft hydrogel (200 μm in thickness) with enhanced surface adhesion was supported on a textile paper backing layer, and then the adhesive surface was used to cover the patterned active array and peel off the device from the glass slide.

### Fabrication of ultrasoft electrophysiology (EP) recording device

For the stretchable conductor-based EP sensors, 200 μL Nafion solution was first mixed with 80 μL DMSO and 5 μL of Triton-100X. Then the mixture was blended with 300 μL PEDOT: PSS solution using a vortex mixer for 30 seconds. The solution was firstly spin-coated on an OTS-treated Si wafer at a speed of 2000 rpm for 1 minute followed by

annealing at 100 °C for 10 minutes. The laser engraving at the power of 6.5% and speed of 10 mm/s was used to pattern the electrode. After removing the redundant area using a tweezer, a SEBS interlayer was then spin-coated (80 mg/mL at 1000 rpm for 1 minute) on top of the patterned electrode surface. Before transferring off the Si wafer, oxygen plasma was applied to the soft interlayer for 1 minute (100 W), and the Dextran sacrificial layer was spin-coated at 2000 rpm for 1 minute on top of the interlayer. Then, a PDMS stamp was used to pick up the interlayer and electrode. Thin copper wires were fixed on the back end of electrodes using silver epoxy (TED PELLA, INC) (15 minutes in an 80 °C oven for curing) for connecting the electrodes with the amplifier. A PET mask (15-μm thick) was then used for screen printing of the top encapsulation layer. The material for top encapsulation was silicon adhesive (Smooth-on), which was painted on the electrode through the PET mask and left three 0.8 mm$^2$ sensing sites exposed. After curing in an 80 °C oven for 10 minutes, a rigid frame made of PET tape was used to peel off the device from the SEBS stamp in DI water, and the device was then laminated to a piece of ultrasoft hydrogel substrate (200 μm). And then a razor blade was used to cut the device into a strip format for wrapping on the heart. Before using the device for isolated heart sensing, the hydrogel was slightly de-swelled in an ambient environment to make it adhesive to the wet surface of the isolated heart.

EP electrodes for wearable ECG recording using the same PEDOT: PSS composite as an electrically conductive layer. PEDOT: PSS layer is patterned into a 1 cm by 1 cm square. For the ultrasoft device, PEDOT: PSS composite is supported on an interlayer (1 μm in thickness) integrated hydrogel substrate (200 μm in thickness); for the high modulus contrast, a 100-μm thick SEBS substrate is used.

### Isolated heart recording

For the ECG measurement on an isolated heart, heart isolation was based on a similar process in a previous report[59]. The LVP was monitored by a BP-100 probe (iWorx), and the electrodes were connected to the preamplifier (C-ISO-256, iworx), and a 4-channel amplifier (IA-400D, iworx) were combined for the EP signal, LVP signal amplification. A DAC system (Digidata 1550B) was used to record the ECG pattern and LVP patterns. The ultrasoft electrodes were wrapped around the isolated heart and held with a clamp that was at the same height as the heart. The EP sensing area was attached near the LV for ECG recordings.

For the ECG measurement on the human skin surface, the electrodes are connected to a commercial ECG measurement kit (Backyard Brains). A notch filer of 60 Hz has been applied, and the ECG signal is measured using two lead modes on the left and right arms.

### In vivo study of immune reactions

Animal: Male C57BL/6 (age 8 weeks) was purchased from Charles River Laboratory. 4–5 mice were housed in a cage, and a sizzle-nest was added in a cage for animal welfare. After the mice were delivered to the animal research center of the University of Chicago, a stabilization period of 7 days was set at 12/12 dark/light cycle (6 am to 6 pm light and 6 pm to 6 am dark), 18–23 degrees Celsius, and 40–60% humidity. The mice's condition was checked by the experimenter twice a day (9 am and 5 pm). All the animal experiments performed in this research were approved by the Institutional Animal Care and Use Committee of the University of Chicago.

Three types of implants have been studied: (1) hydrogel-interlayer bilayer structure; (2) pure PAAm hydrogel; (3) pure SEBS H1052. The PAAm ultrasoft hydrogel was synthesized by dissolving 7.1 g, 100 mmol of acrylamide, 5 mg, 0.032 mmol of crosslinker, 7 mg, 0.031 mol of initiator, and 27 μL, 0.181 mmol TEMED in 50 mL DI water. The prepared solution was injected into a capped PET mold, and fully crosslinked in 80 °C oven for 15 minutes. The hydrogel sheet was then soaked in DI water (12 hours each cycle, changed with fresh water for

three cycles) to wash out the unreacted chemicals. SEBS samples were drop-casted in a glass-based mold to form 200-μm films. For the bilayer structure, firstly, two pieces of SEBS H1052 thin films need to be prepared by spin-coating (60 mg/mL solution at 1000 rpm for 1 min) on an OTS-treated Si wafer. Then, the two thin films were oxygen plasma treated for 1 minute (100 W) and peeled from the substrate with a rigid frame. Using a 5-mm diameter puncher to get an ultrasoft hydrogel disk and put the hydrogel disk in between two SEBS thin films. Slowly decrease the distance between the two thin films and let them adhere to each other forming the encapsulated hydrogel disk. The boundary part can be rolled up to form a better sealing. All three types of samples were sterilized with 70% ethanol solution and UV light for 20 minutes each. Subcutaneous implantation on the back of the animal was accessed by a 1–2 cm skin incision per implant after removing the back hair. Blunt dissection was performed to create the space from the incision towards the animal's shoulder blades. The samples were placed in the subcutaneous pocket created above the incision, and the incision was closed with interrupted sutures. The animals were sacrificed after 1-month by $CO_2$ asphyxiation.

### Histological analysis (trichrome staining)

The mice skin samples were harvested on the scheduled date and further incubated in 2% PFA for 2 days (4 °C). After fixation of the skin samples, they were embedded in paraffin and sectioned at a thickness of 5 μm. The Human Tissue Resource Center at The University of Chicago carried out the Trichrome staining, and the slides were imaged with EVOS EL Auto (Life Technologies).

### Immunofluorescence analysis

The skin samples were first performed deparaffination process. And the slides were incubated in perm/blocking buffer (0.3% Triton X-100, 1% BSA in PBS) for 3 hours at room temperature. The slides were then washed in 1× PBS three times and incubated in primary antibody solution (0.1% tween 20 in PBS) for overnight (4 °C). The slides were washed using 1× PBS three times and incubated in secondary antibody solution (0.1% tween 20 in PBS) for 3 hours at room temperature. Secondary antibody information: Goat anti-Rat IgM (Heavy chain) Secondary Antibody Alexa Fluor™ 647 and Donkey anti-Rabbit IgG (H + L) Highly cross-adsorbed Secondary Antibody, Alexa Fluor™ 594 (Invitrogen). Then, the slides were washed using 1× PBS three times and stained using DAPI. The stained skin slides were covered with mounting solution and dried overnight in dark place. The slides were imaged by Olympus confocal microscopy system. The following antibodies were used for immunofluorescence staining: smooth muscle actin Polyclonal antibody (Proteintech, Cat# 14395-1-AP, Lot# 00081698, clone: polyclonal, dilution: 1:800), CD68 Monoclonal Antibody (Invitrogen, Cat# 14-0681-82, Lot# 2472523, clone: FA-11, dilution: 1:100), Donkey anti-Rabbit IgG (H + L) Highly cross-adsorbed Secondary Antibody, Alexa Fluor™ 594 (Invitrogen, Cat# A-21207, Lot# 2145022, clone: polyclonal, dilution: 1:500), Goat anti-Rat IgM (Heavy chain) Secondary Antibody Alexa Fluor™ 647 (Invitrogen, Cat# A21248, Lot# 1964390, clone: polyclonal, dilution: 1:500).

### Finite-element simulations

Finite-element software COMSOL Multiphysics was used to model a three-layer structure subjected to external uniaxial strain and map the stress level. To investigate the concentrated stress that may lead to a crack channeling across the thin film, we built a three-dimensional model. The modulus of the top thin film was 19.4 MPa (modulus of the nanoconfinement DPP). The hyper-elastic material model (Neo-Hookean) was applied to the functional thin film, soft interlayer (modulus of the SEBS H1052, 2.83 MPa), and the ultrasoft substrate (modulus of the ultrasoft hydrogel, 2.5 kPa). We varied the modulus (10 kPa to 1 GPa) and thickness (100 nm to 10 μm) of the soft interlayer, and the delaminated area (from 1.5 to 41 times to the notch area) between the

functional layer and the interlayer around the notch, and calculated the stress level. To simplify the simulation, we assume the adhesion between the layers is infinite, and no extra delamination occurs in the three-layer structure under the uniaxial stretching (up to 10% strain). Hexahedral elements (with a minimal element size of 0.8 nm) were used to mesh the structure.

### Reporting summary

Further information on research design is available in the Nature Portfolio Reporting Summary linked to this article.

## Data availability

The authors declare that all data supporting the findings of this study are available within the paper and its Supplementary Information files or available from the corresponding authors upon request. Source data are provided as a Source Data file. Source data are provided with this paper.

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

## Acknowledgements

This research is supported by the US National Institutes of Health Director's New Innovator Award 1DP2EB034563, the US Office of Naval Research Young Investigator Award N00014-21-1-2581, the University of Chicago Materials Research Science and Engineering Center, which is funded by the National Science Foundation under award number DMR-2011854, and the start-up funds from the University of Chicago. This research also used the Soft Matter Characterization Facility (SMCF) at the University of Chicago.

## Author contributions

Ya.L. and Si.W. conceived the concept and designed the experiments. N.L. synthesized the polymer semiconductor. W.L. developed the polymer conductor. A.P. and Z.C. performed isolated heart experiments. S.K., Sh.W., and N.L. worked on immune

studies. Y.D. and Yo.L. helped with hydrogel device fabrication. H.H. and S.D. helped with semiconductor characterization and analysis. Q.S. and P.C. helped with hydrogel device characterization. C.W. and L.J. provided guidance on mechanical simulations and edited the manuscript. J.H. and B.T. provided comments for the manuscript. Ya.L. and Si.W. wrote the paper.

## Competing interests

The authors declare no competing interests.
