## [Peer Review File · Nature Communications]

Achieving tissue-level softness on stretchable electronics through a generalizable soft interlayer designREVIEWER COMMENTS

Reviewer #1 (Remarks to the Author):

This study presents stretchable transistor devices fabricated on a soft hydrogel substrate, as opposed to a stretchable elastomer substrate such as SEBS. The authors applied a thin layer of SEBS on the hydrogel substrate to create stretchable devices, enabling the use of highly soft hydrogel-based elastomers. The authors aimed to amplify the advantages of using a soft substrate by attaching the device to a beating heart and confirming stable attachment without any position movement of the device. Previous articles have investigated the effect of gradual modulus design on improving the stretchability of electrodes. The main contribution of this work is the design rule for optimizing stretchability with the addition of an interlayer. I recommend publication of this work, but before publication, the authors should address the following comments.

1. The stiffness of an elastomer substrate depends on its thickness, and fabricating a device directly on a thin SEBS substrate could result in an ultrasoft device. The comparison demonstration presented in this work may appear unfair, thus the authors should address this point to provide a more convincing argument.
2. This work claims that the interlayer functions as a water passivation layer, but it is unclear whether the soft interlayer can prevent water penetration and prevent damage to the transistor's performance. The authors should confirm the effectiveness of the water passivation layer.
3. The authors should provide more details on the design of the interlayer to provide more general information on its application.
4. The demonstration on heart is not convincing yet because the stable attachment may not owing to the softness but to the presence of hydrogel.

Reviewer #2 (Remarks to the Author):

In this manuscript, Li et al introduce a general soft interlayer design that can enable stretchable materials with relatively high moduli to be utilized for developing stretchable electronics. Assisted by the interlayer design, the uniform strain distribution could achieve the stretchability of a functional layer with a much lower modulus. Such tissue-like modulus shows great advantages with good conformability for irregular surfaces, low mechanical constraint, and reliable working capability. The overall work is interesting and has potential in stretchable electronics and this work can be published after addressing the following comments.

1. For the SEBS interlayer, how to control the thickness precisely?

2. The authors mainly used SEBS as the interlayer, while there are multiple stretchable elastomers with similar modulus, how about the performance of other stretchable materials as the interlayer?
3. To develop stretchable devices, either intrinsically stretchable materials or induced stretchable architectures have been widely discussed. The author should summarize the related works to demonstrate the advantages and disadvantages of this work.
4. SEBS is elastic and hard to fully recover after a large strain. How about the deformation or shape variation of the whole device after large stretching strain?
5. For the ex-vivo test, the authors claimed that the capillary force between the heart and the device helps to provide necessary adhesion to avoid shifting or rotation during the heartbeat. I am curious that for the real in-vivo test, there are pericardial fluids surrounding the heart, how to make sure long-term adhesion and working performance? Besides, how does the adhesion after multiple attaching-releasing processes?
6. How about wearable ECG monitoring with these EP electrodes?
7. The authors should check the figures carefully and add the corresponding scale bars for all the images, such as Fig 3a, 3c, 3d, 3g, S1, and others.
8. For Fig 3a, what is the strain condition for the transistors, and how about the surface characterization under strain?

Reviewer #3 (Remarks to the Author):

The authors describe an interesting approach to design conformable soft tissue-compatible bioelectronic sensing devices by using an interlayer design. They fabricated stretchable transistor arrays on their SEBS-layered hydrogel substrate and achieved an impressive moduli of less than 10KPa. Further, they demonstrated the in vivo compatibility of their soft device with live rat heart and showed the promising results of measurement of contractile beating events. I recommend publication after addressing the following issues:

(1) are the thickness for ultrasoft device and conventional device in Figure 4g the same? If so, please add such information into the caption.

(2) when comparing the quality of ECG signals of their ultrasoft electrode versus SEBS-based electrode (Supplementary Figs. 24 and 25), it appears that SEBS-based system offered better signal quality. can they explain this?

(3) the underlying science appears to be due to the application of interlayer materials with moduli between softer hydrogen and harder semiconductor. if this is the case, I wonder if other elastomers may be used beyond SEBS. the gradient soft/hard interface design appears to be the key for their device performance, for which a related review article may be cited in the context of their work (Advanced Materials 2020, 32 (15), 1902278).

(4) in figure 1F, a serious delimitation was observed for the conventional devices. However, it was really representative since there are some exemplar elastomer only based soft electronic devices and tattoos that are able to survive in the similar bending finger conditions.

Response letter

Reviewer #1:

This study presents stretchable transistor devices fabricated on a soft hydrogel substrate, as opposed to a stretchable elastomer substrate such as SEBS. The authors applied a thin layer of SEBS on the hydrogel substrate to create stretchable devices, enabling the use of highly soft hydrogel-based elastomers. The authors aimed to amplify the advantages of using a soft substrate by attaching the device to a beating heart and confirming stable attachment without any position movement of the device. Previous articles have investigated the effect of gradual modulus design on improving the stretchability of electrodes. The main contribution of this work is the design rule for optimizing stretchability with the addition of an interlayer. I recommend publication of this work, but before publication, the authors should address the following comments.

Response: We appreciate the positive comments from the reviewer.

1. The stiffness of an elastomer substrate depends on its thickness, and fabricating a device directly on a thin SEBS substrate could result in an ultrasoft device. The comparison demonstration presented in this work may appear unfair, thus the authors should address this point to provide a more convincing argument.

Response: Thanks to the reviewer for the constructive suggestion. Indeed, using a thin SEBS substrate can also help to reduce the stiffness of the entire device, which will benefit their adhesion, conformability, and mechanical constraint to soft tissues. In our case, a thickness of 1 μm for the SEBS substrate could achieve comparable stiffness with our hydrogel device. However, in such a low thickness, the device could become very easy to break (i.e., with low mechanical robustness) and have very low self-supporting ability (i.e., difficult to handle)¹. Therefore, one of the benefits of our interlayer and ultrasoft substrate design is to maintain good mechanical durability while achieving very low stiffness for better conformability.

To provide a more convincing argument regarding this in the manuscript, we have added the following sentence on Page 5, as “**It should be noted that although reducing thicknesses without changing the modulus could also help to decrease the stiffness and achieve better conformability, this comes with the price of sacrificing mechanical robustness.**”

2. This work claims that the interlayer functions as a water passivation layer, but it is unclear whether the soft interlayer can prevent water penetration and prevent damage to the transistor's performance. The authors should confirm the effectiveness of the water passivation layer.

Response: Thanks to the reviewer for the constructive suggestion. To more rigorously show that our interlayer can serve as the water passivation layer, we further performed the electrochemical impedance spectroscopy (EIS) of such a SEBS layer (with the same thickness as the interlayer) coated on a gold electrode. As shown in the figure below, the test in 1 \times PBS solution with a frequency range from 0.1 Hz to 1 MHz indicates a charge transfer impedance of 36 M Ω (fitted using Randles circuit model)², which suggests effective water-passivation and insulating properties of SEBS interlayer for the gold pad. To address such property, this sentence has been added to the manuscript (on page 9): “**which also provides electrical encapsulation for the device (Supplementary Fig. 13)**”

Supplementary Fig. 13. **a**, Electrochemical impedance spectroscopy (EIS) of SEBS interlayer in $1\times$ PBS solutions. The inserted schematic shows the measurement setup of EIS. **b**, Nyquist plot from the measured result, and fitting curve using Randles model (equivalent circuit inserted) is also plotted.

3. The authors should provide more details on the design of the interlayer to provide more general information on its application.

Response: Thanks for the suggestion. In the manuscript, we have proposed three main design aspects for soft interlayers: 1) modulus mismatch, 2) interlayer thickness and 3) interfacial toughness. The detailed trends of their influences have been studied through the finite-element simulations shown in Fig. 2a-c. To give the perspectives on this aspect, following contents are added to the manuscript (on page 8):

“According to such simulation results, the three design aspects should meet the following semi-quantitative relationships with the parameters of the functional layers: 1) a modulus within three orders of magnitude to

the functional layer; 2) adhesion with an interfacial toughness larger than $100 \text{ J}\cdot\text{m}^{-2}$; 3) thickness at least ten times thicker than the functional layer.”

Supplementary Table 1 Details of interlayer design for electrically functional materials^{3,4}.

Functional materials (interlayer materials)	Modulus mismatch	Interfacial toughness	Optimized thickness	Stretchability
PEDOT: PSS & PFI (SEBS)	1.05 GPa (2.83 MPa)	$14.2 \text{ J}\cdot\text{m}^{-2}$	$>10 \mu\text{m}$	$>80\%$
DPPT-TT & SEBS (SEBS)	1.94 MPa (2.83 MPa)	$402.7 \text{ J}\cdot\text{m}^{-2}$	$>1 \mu\text{m}$	$>100\%$
CNT (SEBS)	4.1 GPa (2.83 MPa)	$178.8 \text{ J}\cdot\text{m}^{-2}$	$>0.5 \mu\text{m}$	$>200\%$
AgNWs (TPU)	0.88 GPa (9.6 MPa)	$269.7 \text{ J}\cdot\text{m}^{-2}$	$>0.5 \mu\text{m}$	$>180\%$

4. The demonstration on heart is not convincing yet because the stable attachment may not owing to the softness but to the presence of hydrogel.

Response: We highly appreciate this comment from the reviewer. First, to clarify, the integration of our devices (both the hydrogel one and SEBS one) to the heart is not mainly through the direct adhesion to the heart surface. Rather, the devices were wrapped around the heart, as a ring structure, so that we can control the tightness of the device wrapping and use the capillary force to keep the devices on the heart. Second, the PAAm hydrogel we prepared and utilized as the substrate for our device actually also has relatively poor adhesion with the wet heart tissue. To confirm this, we further used standard lap shearing tests to compare the shear strengths of the hydrogel and SEBS layer attaching to heart tissue, in reference to a commercial tissue adhesive (BioGlue), based on the shear strength value reported in a previous paper⁵. The results (as shown below) have been added into the **Supplementary Information as Fig. 27**. We can find that the hydrogel and SEBS indeed both have much lower adhesion than the tissue adhesive. So overall, without a strong adhesion of the devices on the heart surface, the low modulus of the ultrasoft device, compared to the SEBS devices, enables easier deformation of the device to accommodate the heart beating, thereby helping to reduce the interfacial shear stress. This is also reflected by the much less increase of the ventricle pressure from the attachment of the ultrasoft device. So overall, we think in this demonstration setting, the stable attachment of the device on the heart should be mainly benefited from the decreased modulus of the device. To address this aspect, related text in the manuscript (page 15) has been revised as follows:

“This EP device is attached to the heart surface by wrapping it as a circular ring. Due to the existence of a buffer solution, capillary force formed between the heart and device would provide necessary adhesion. (Supplementary Fig. 27)”

Supplementary Fig. 27. Adhesion between SEBS, PAAm hydrogel, and a commercial adhesive (BioGlue) on wet heart tissue, as measured by lap shear test. **a**, Stress-displacement curves for PAAm hydrogel and SEBS. **b**, Shear strength of PAAm hydrogel, SEBS, and a commercial adhesive (with the value extracted from ref. R5).

Reviewer #2:

In this manuscript, Li et al introduce a general soft interlayer design that can enable stretchable materials with relatively high moduli to be utilized for developing stretchable electronics. Assisted by the interlayer design, the uniform strain distribution could achieve the stretchability of a functional layer with a much lower modulus. Such tissue-like modulus shows great advantages with good conformability for irregular surfaces, low mechanical constraint, and reliable working capability. The overall work is interesting and has potential in stretchable electronics and this work can be published after addressing the following comments.

Response: We sincerely thank the reviewer for the positive comments and suggestions.

1. For the SEBS interlayer, how to control the thickness precisely?

Response: Thanks a lot for this question from the reviewer. The SEBS interlayer can be prepared by spin-coating or drop-casting on a sacrificial layer coated glass slide. For the spin-coating process, the thickness of the obtained film is mainly determined by the concentration of the solution and the spin rate. To build a more quantitative relationship that other researchers can use directly, we further measured the film thicknesses obtained from four concentrations between 30 mg/mL and 100 mg/mL, and five spin rates between 1,000 rpm and 6,000 rpm. The results are shown in the newly added Supplementary Fig. 31, and also copied below. For getting films thicker than 20 μm , drop casting of high-concentration SEBS solutions (150 mg/mL to 200 mg/mL) can be used, for which the film thickness can be controlled by both the concentration and the total amount of solution cast on a certain surface area. Preparation of SEBS interlayer has been added to the Methods section (page 18): “The correlation between the thicknesses of SEBS interlayer and the spin-coating rates has been summarized in Supplementary Fig. 31.”

Supplementary Fig. 31. Relationship between SEBS interlayer thickness and spin-coating rate. Four concentrations are used to generate SEBS interlayers ranging from 140 nm to 3.5 μm .

2. The authors mainly used SEBS as the interlayer, while there are multiple stretchable elastomers with similar modulus, how about the performance of other stretchable materials as the interlayer?

Response: Thanks for the comments from the reviewer. Based on the theory of crack propagation in thin film, we choose the materials with high adhesion to the functional materials to serve as the interlayer. For

polymer semiconducting films (with DPPT-TT/SEBS blends as an example) and stretchable CNT electrodes, SEBS—compared to other typical elastomers such as PDMS and polyurethane—gives the best adhesion. To further compare the difference of these additional elastomers as the interlayer, we characterized the stretchability of polymer semiconductor films on these layers of 1- μm thick, all with Ecoflex as the substrate. During stretching to 100% strain, the crack propagation in each case is shown by optical microscopy images (see figures below). Although TPU and PDMS have similar modulus (9.56 MPa and 1.85 MPa, respectively) to SEBS (2.83 MPa), the adhesions to the semiconductor films are different. Therefore, larger crack sizes are observed on the PDMS interlayer supported semiconductor films compared to the SEBS interlayer or TPU interlayer. For the TPU interlayer, it has slightly lower adhesion to PSC films than SEBS, but the higher Young's modulus than SEBS also delays the cracks propagation. Overall, TPU interlayer shows a similar buffering effect to the SEBS interlayer. However, to reduce the overall modulus of the device, SEBS interlayer is preferred.

Supplementary Fig. 5. Morphology of CONPHINE films supported by interlayers made of **a**, SEBS, **b**, TPU and **c**, PDMS, during stretching to 100% strain. SEBS and TPU interlayers have similar mechanical buffering effects for delayed crack propagation in CONPHINE films. For the PDMS interlayer, obvious cracks formed under 25% strain, and cracks at much larger scale are observed under 100% strain than the SEBS and TPU interlayers. Scale bars, 20 μm .

3. To develop stretchable devices, either intrinsically stretchable materials or induced stretchable architectures have been widely discussed. The author should summarize the related works to demonstrate the advantages and disadvantages of this work.

Response: We highly appreciate this constructive suggestion from the reviewer. In the manuscript, Fig. 3h summarizes the comparison of our ultrasoft stretchable transistors with the representative works of previous

reported intrinsically stretchable transistors and passive electrode-type devices, in terms of the stretchability and moduli. Following the suggestion from the reviewer to make such comparison clearer, we have added a supplementary table (Supplementary Table 2, also copied below) to detail the comparison, also adding information about the substrate materials, functional layer designs, etc. Compared to these reported devices, the main advantages provided by our soft interlayer design are to achieve a decrease of device moduli by 3-order-of-magnitude or more, without sacrificing stretchability, while still realizing advanced transistor performance. When compared with stretchable devices using inorganic electronic materials based on strain engineering, the advantage of tissue-like softness of our ultrasoft polymer-based devices is even more prominent, especially for the local moduli at the device locations.

Supplementary Table 2 Details of interlayer design for electrically functional materials.

Key design feature	Stretchability	Substrate modulus	Device type	Reference
Au coated polyurethane nanomesh	30%	0.274 MPa	Passive	11
PEDOT: PSS hydrogel	20%	30 kPa	Passive	29
PAAm-PEDOT: PSS hydrogel	N.A.	2.5-4.0 GPa (polyimide)	Passive	34
Ag flakes in PAAm-alginate hydrogel	250%	3-4 kPa	Passive	36
DPPT-TT & SEBS composite	100%	1 MPa	Active	37
N2200 with PU encapsulation	50%	0.8~2.5 MPa (PDMS)	Active	38
P3HT-NFs & PDMS composite	30%	0.7 MPa	Active	48
Soft interlayer	100%	5.2 kPa	Active	Our work

4. SEBS is elastic and hard to fully recover after a large strain. How about the deformation or shape variation of the whole device after large stretching strain?

Response: Thanks for the valuable questions. Indeed, SEBS has viscoelastic behavior and cannot immediately return to its original length after large deformation. However, since the SEBS interlayer is made with a thickness of 1 μm , which is much thinner compared to the hydrogel substrate of 200 μm in thickness, so the overall mechanical behavior of the device is dictated by the ultrasoft hydrogel substrate. To show the hydrogel dictated elastic property in the ultrasoft transistor device, the device is stretched to 100% strain and released (as shown in Fig. R1). The ultrasoft transistor device fully recovered from the 100% strain to its original length, and overall, the device shows no shape variation after large stretching strain.

Fig. R1. **a**, unstretched transistor device. **b**, stretching the device to 100% strain. **c**, transistor device recovered from 100% strain.

5. For the ex-vivo test, the authors claimed that the capillary force between the heart and the device helps to provide necessary adhesion to avoid shifting or rotation during the heartbeat. I am curious that for the real in-vivo test, there are pericardial fluids surrounding the heart, how to make sure long-term adhesion and working performance? Besides, how does the adhesion after multiple attaching-releasing processes?

Response: Thanks for the valuable questions from the reviewer. We fully agree with the reviewer that in-vivo uses generally pose a higher level of challenge for achieving stable adhesion of bioelectronic devices on tissue/organ surfaces, exactly due to the reasons stated by the reviewer. Overall, the adhesion stability is determined by the competition between two factors—adhesion force at the interface and shear stress caused by strain mismatch. This work mainly focuses on the device designs for minimizing the shear stress, as it is proportional to the stiffness of the attached device. The first aspect of adhesive properties at the tissue interface is not a part of the focus of this work. We envision that this aspect can be significantly improved by using tissue-adhesive elastomers/hydrogels as device substrates, and tissue-adhesive conducting polymers as the sensing surface. From recent literature^{5,6}, such tissue-adhesive materials with promising performance are already readily available for use in such ultrasoft bioelectronic devices. For this, our soft interlayer design is particularly beneficial owing to its universal applicability to different materials. Based on the reported behaviors of some of such tissue-adhesive materials^{7,8}, adhesion properties can be well maintained after multiple cycles of attaching-releasing.

6. How about wearable ECG monitoring with these EP electrodes?

Response: Thanks to the reviewer for this question. To answer this question, we further carried out a set of experiments of using our ultrasoft EP electrode device (in comparison to a SEBS-based EP device) for wearable ECG monitoring. Stretchable PEDOT: PSS electrodes in same dimension (square of 1 cm sides) are supported with SEBS substrate or interlayer with ultrasoft hydrogel substrate for comparison. SEBS substrate has a thickness of 100 μm , and the hydrogel substrate is 200 μm in thickness with a 10 μm SEBS interlayer. ECG is recorded using a commercially available ECG kit (Backyard Brains), with the two electrodes attached to the left and right arms. The results are added to the manuscript as **Supplementary Fig. 30**, which is also copied below. The device based on the SEBS substrate shows poor contact with the skin and motion artifacts causing distorted signals. For the ultrasoft device with the soft interlayer design, it

remains in good contact when the arm is moving, and the recorded ECG signal is stable and in higher amplitude. A sentence is added to the manuscript (page 16) to demonstrate the advantage of ultrasoft device as wearable ECG sensor: “Such stable EP recording has also been achieved on the human skin surface with a similar design (Supplementary Fig. 30)”

Supplementary Fig. 30. Stretchable PEDOT: PSS electrodes for ECG recording on skin surface. **a**, electrodes based on SEBS interlayer and ultrasoft hydrogel substrates. The low effective modulus enables good contact with skin and remains stable for the ECG recording. **b**, stretchable electrode based on SEBS substrate is less conformal and results in recorded signal with lower amplitude. Also, the high stiffness makes it hard to remain intimate contact with skin when the arm is moving, which further results in a distorted signal.

7. The authors should check the figures carefully and add the corresponding scale bars for all the images, such as Fig 3a, 3c, 3d, 3g, S1, and others.

Response: Thanks very much for the suggestion. Scale bars have been added to the images.

8. For Fig 3a, what is the strain condition for the transistors, and how about the surface characterization under strain?

Response: We highly appreciate the questions from the reviewer. The strain applied to the transistor device is 300%. To characterize the transistor surface under strain, we use confocal microscopy imaging. As shown in the newly added **Supplementary Fig. 11**. (copied below), under 300% strain, wrinkled structures are found on the transistor area, which is because of the modulus mismatch between transistor layers and the ultrasoft hydrogel substrate⁹. The amplitude of the wrinkling structure is around 10 μm .

Supplementary Fig. 11. Confocal microscope imaging of a strained ultrasoft transistor device. Wrinkled structures are observed on the surface due to the modulus mismatch between functional layers and the ultrasoft hydrogel substrate (scale bar, 500 μm).

Reviewer #3:

The authors describe an interesting approach to design conformable soft tissue-compatible bioelectronic sensing devices by using an interlayer design. They fabricated stretchable transistor arrays on their SEBS-layered hydrogel substrate and achieved an impressive moduli of less than 10KPa. Further, they demonstrated the in vivo compatibility of their soft device with live rat heart and showed the promising results of measurement of contractile beating events. I recommend publication after addressing the following issues:

Response: We really appreciate the positive comments from the reviewer.

(1) are the thickness for ultrasoft device and conventional device in Figure 4g the same? If so, please add such information into the caption.

Response: Thanks the reviewer for the suggestion. The thicknesses for ultrasoft device and conventional device are different. The thickness of ultrasoft device is 200 μm and the conventional device is 100 μm . Information has been added to the manuscript (page 5)

(2) when comparing the quality of ECG signals of their ultrasoft electrode versus SEBS-based electrode (Supplementary Figs. 24 and 25), it appears that SEBS-based system offered better signal quality. can they explain this?

Response: We thank the reviewer for the question. To more accurately compare the signal quality, we further calculated signal-to-noise ratios (SNR) for the recorded signals from both ultrasoft devices and SEBS supported devices. The signal from the ultrasoft device has an SNR of 19.65 dB, while the SEBS-based device has an SNR of 16.73 dB. Therefore, there are no obvious differences in the signal quality. The SNR is calculated using the following equation:

$$\text{SNR} = 20 \log_{10} (V_{\text{signal}}/V_{\text{noise}})$$

SNR calculation of the EP devices have been added to the manuscript in Supplementary Information Figure 28 and 29.

(3) the underlying science appears to be due to the application of interlayer materials with moduli between softer hydrogen and harder semiconductor. if this is the case, I wonder if other elastomers may be used beyond SEBS. the gradient soft/hard interface design appears to be the key for their device performance, for which a related review article may be cited in the context of their work (Advanced Materials 2020, 32 (15), 1902278).

Response: We really appreciate this suggestion from the reviewer. This reference has been added to the manuscript on page 2.

(4) in figure 1F, a serious delimitation was observed for the conventional devices. However, it was really representative since there are some exemplar elastomer only based soft electronic devices and tattoos that are able to survive in the similar bending finger conditions.

Response: We thank the reviewer for this highly insightful suggestion. To achieve a high conformability and remain stable on skin surface, several approaches can be applied to stretchable devices, such as a)

reducing modulus, b) enhance the adhesion^{10–12} and c) reducing the thickness^{13,14}. Since thin-film devices with high conformability have been achieved in the reported results, here, we are exploring new form-factor for enhancing conformability to the skin surface. Our interlayer and ultrasoft substrate strategy doesn't rely on decreasing the device's thickness but on reducing the modulus to achieve high conformability on dynamic skin surface. To address the effectiveness of reducing modulus, the SEBS device and ultrasoft device are prepared with similar thicknesses (100 μm and 200 μm , respectively).

References

- R1. Missale, E., Frascioni, M. & Pantano, M. F. Ultrathin organic membranes: Can they sustain the quest for mechanically robust device applications? *iScience* **26**, 105924 (2023).
- R2. Floch, P. L. *et al.* Fundamental Limits to the Electrochemical Impedance Stability of Dielectric Elastomers in Bioelectronics. *Nano Lett* **20**, 224–233 (2019).
- R3. Li, X., Gao, H., Murphy, C. J. & Caswell, K. K. Nanoindentation of Silver Nanowires. *Nano Lett.* **3**, 1495–1498 (2003).
- R4. Chen, L. *et al.* Auxetic materials with large negative Poisson's ratios based on highly oriented carbon nanotube structures. *Appl. Phys. Lett.* **94**, 253111 (2009).
- R5. Yuk, H. *et al.* Dry double-sided tape for adhesion of wet tissues and devices. *Nature* **575**, 169–174 (2019).
- R6. Li, J. *et al.* Tough adhesives for diverse wet surfaces. *Science* **357**, 378–381 (2017).
- R7. Deng, J. *et al.* Electrical bioadhesive interface for bioelectronics. *Nat. Mater.* **20**, 229–236 (2021).
- R8. Yang, Q. *et al.* Photocurable bioresorbable adhesives as functional interfaces between flexible bioelectronic devices and soft biological tissues. *Nat. Mater.* 1–12 (2021) doi:10.1038/s41563-021-01051-x.
- R9. Stafford, C. M. *et al.* A buckling-based metrology for measuring the elastic moduli of polymeric thin films. *Nat. Mater.* **3**, 545–550 (2004).
- R10. Drotlef, D.-M., Amjadi, M., Yunusa, M. & Sitti, M. Bioinspired Composite Microfibers for Skin Adhesion and Signal Amplification of Wearable Sensors. *Adv. Mater.* **29**, 1701353 (2017).
- R11. Min, H. *et al.* Tough Carbon Nanotube-Implanted Bioinspired Three-Dimensional Electrical Adhesive for Isotropically Stretchable Water-Repellent Bioelectronics. *Adv. Funct. Mater.* **32**, 2107285 (2022).
- R12. Wang, X. *et al.* Bioinspired Interface-Guided Conformal Janus Membranes with Enhanced Adhesion for Flexible Multifunctional Electronics. *Chem. Mater.* **34**, 5980–5990 (2022).
- R13. Wang, Y. *et al.* A durable nanomesh on-skin strain gauge for natural skin motion monitoring with minimum mechanical constraints. *Sci. Adv.* **6**, eabb7043 (2020).
- R14. Lee, S. *et al.* Nanomesh pressure sensor for monitoring finger manipulation without sensory interference. *Science* **370**, 966–970 (2020).

REVIEWERS' COMMENTS

Reviewer #1 (Remarks to the Author):

The authors addressed well the comments from the reviewers. I recommend publication of this work.

Reviewer #2 (Remarks to the Author):

The authors have addressed my previous comments. Now the paper is suitable for publication.

Reviewer #3 (Remarks to the Author):

the authors have addressed all my comments. it is now publishable